# Agent Aggregator with Mask Denoise Mechanism for Histopathology Whole Slide Image Analysis

## ABSTRACT

Histopathology analysis is the gold standard for medical diagnosis. Accurate classification of whole slide images (WSIs) and region-of-interests (ROIs) level localization will assist pathologists in clinical diagnosis. With a gigapixel resolution and a scarcity of fine-grained annotations, WSI is difficult to classify directly. In the field of weakly supervised learning, multiple instance learning (MIL) serves as a promising approach to solving WSI classification tasks. Currently, a prevailing aggregation strategy is to apply attention mechanism as a measure of the importance of each instance for further classification. Notwithstanding, attention mechanism fails to capture inter-instance information and self-attention mechanism can cause quadratic computational complexity issues. To address these challenges, we propose an agent aggregator with mask denoise mechanism for multiple instance learning termed AMD-MIL. The agent token represents an intermediate variable between the query and key for implicit computation of the instance importance. Mask and denoising are also learnable matrices mapped from the agents-aggregated value, which first dynamically mask out some low-contribution instance representations and then eliminate the relative noise introduced during the mask process. AMD-MIL can indirectly achieve more reasonable attention allocation by adjusting feature representations, thereby sensitively capturing micro-metastases in cancer and achieving better interpretability. Our extensive experiments on CAMELYON-16, CAMELYON-17, TCGA-KIDNEY, and TCGA-LUNG datasets show our method's superiority over existing state-of-the-art approaches. The code will be available upon acceptance.

## CCS CONCEPTS

• **Computing methodologies** → **Computer vision tasks**.

## KEYWORDS

histopathology diagnosis, multiple instance learning, agent attention, mask denoise mechanism

## 1 INTRODUCTION

The advancement of deep learning technologies and increased computational capacities have significantly enhanced the field of computational pathology [2, 12, 16, 23]. This progress assists physicians

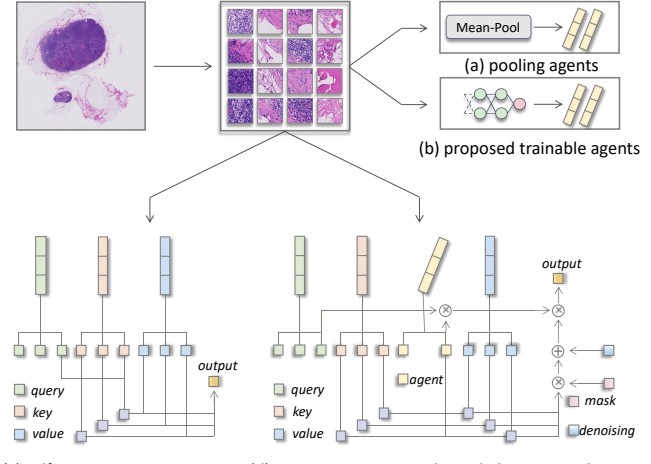

**Figure 1: Comparison of core modules: (a) pooling agents. (b) proposed trainable agnets. (c) self-attention mechanism. (d) proposed agent aggregator with mask denoise mechanism. Mask and denoising are learnable matrices.**

in diagnosis and standardizes pathological diagnostics [7, 20]. However, analyzing histopathology whole slide images (WSIs) markedly differs from typical computer vision tasks [19]. A single WSI, with its gigapixel resolution, makes obtaining pixel-level annotations impracticable, in contrast to natural images [6]. The Multiple Instance Learning (MIL) approach is currently the mainstream framework for analyzing pathology slides using only WSI-level annotations [18, 24, 31]. This method considers the entire WSI as a bag, with each patch within it as an instance [4, 29]. If any instance within the WSI is classified as cancerous, then the entire WSI is labeled as such [3, 28]. The WSI is labeled as normal only if all instances within it are normal.

Current MIL methodologies are typically divided into two stages. Initially, the entire WSI is segmented into numerous patches. After segmentation, a pre-trained feature extractor is used to embed features in each patch, creating a comprehensive representation of the WSI. The subsequent stage involves aggregating these features. Various mechanisms, from standard pooling like mean-pooling and max-pooling to advanced attention-based methods like ABMIL [11], DSMIL [13], and TransMIL [21], are employed in this process. These aggregated features are then mapped to categorical dimensions for classification purposes.

ABMIL [11] employs the standard attention mechanism for information aggregation. However, due to the lack of consideration for the relationships between different instances, this kind of method struggles with global modeling and long-distance dependency capturing. TransMIL and MMIL [35] use self-attention [27] within

MIL's feature aggregator. Self-attention can calculate the relations between any two patches within a WSI, thus enabling the capture of long-distance dependencies within the image. Moreover, self-attention dynamically allocates weights according to the importance of the input elements, enhancing the model's ability to process complex data. Nonetheless, due to the quadratic complexity of self-attention, its real-world application within MIL aggregators is challenging.

To address the quadratic complexity issue of self-attention, Trans-MIL [21] employs nystrom attention [32] as the substitute for the standard self-attention module. Nystrom attention selects a subset of sequence elements, also known as landmarks, to approximate the attention scores for the entire sequence. Specifically, in the nystrom attention mechanism, the local downsampling of query and key matrices is implemented along the dimension of the instance tokens. This approach has two significant issues. Firstly, since the sampling process relies on adjacent instances, many insignificant ones might dilute the impact of significant instances. Secondly, Equidistant division is not always the optimal sampling strategy, as the distribution of information in a sequence may be uneven. Fixed sampling intervals might fail to capture all crucial information points, leading to a decrease in approximation quality.

To address these challenges, We transform the pooling agent within the agent attention [9] mechanism into trainable matrices, aiming for an effective agent mapping. Furthermore, to indirectly achieve a more rational distribution of attention scores through adjustments in instance representations, we introduce mask denoise mechanism for dynamic adaptation.

Agent attention [9] introduce the agent tokens in addition to the query, key, and value tokens. Agent token acts as an agent for the query tokens, aggregating information from the key and value tokens, and then information is returned to the query tokens via a broadcasting mechanism. Given the lesser number of agent tokens compared to sequence tokens, agent mechanism can reduce the computational load of standard self-attention. However, agent tokens are obtained through mean pooling of the query tokens in standard agent attention, making it challenging to adapt to the variable-length token inputs of pathological multiple instance tasks. Additionally, mean pooling, by aggregating features through local averaging, fails to capture some long-instance dependencies. Consequently, we adjust the number of agent tokens as a hyperparameter and substitute the mean pooling agent tokens with trainable agent tokens.

Moreover, we introduce a mask denoise mechanism to dynamically refine attention scores by adjusting instance representations. Mask and denoising matrices, matching the agent's aggregated value dimension, are generated by projecting this value through a linear layer. Mask matrices transform into binary matrices via threshold filtering, not directly from the value token but its high-level mapping, allowing dynamic adaptation to the input. Then, the mask directly multiplies with the value, filtering out non-significant features. However, as the mask applies binary filtering to the value, it might suppress unimportant instances excessively, thereby introducing relative noises. Therefore, we introduce the denoising matrices from the agent-aggregated values to correct the relative

noises. We conducted extensive comparative experiments and ablation studies on four datasets to verify the effectiveness of the trainable agent aggregator and mask denoise mechanism.

## 2 RELATED WORK

### 2.1 Multiple Instance Learning for WSI Analysis

MIL methodology demonstrates significant potential in the classification and analysis of pathological images. In this framework, a WSI is treated as a bag and the local regions within it are considered instances. Primarily, MIL paradigms are categorized into three types: instance-based, embedding-based, and bag-based methods. The instance-based method scores each instance, and then aggregates these scores to predict the bag's label. The embedding-based method initially uses a pre-trained feature encoder to obtain instance representations, then aggregates them for classification. Instance representations share the same feature space, enhancing fit with Deep Neural Networks (DNN) but reducing interpretability. Bag-based approaches classify by comparing distances between different bags, with the main challenge being to identify a universal distance metric for global comparison. Current advancements in MIL methodology focus on the development of specialized feature encoders pre-trained on pathological datasets, enhancements in aggregator techniques, augmentation of training data, and the improvement of training strategies.

Feature encoders pre-trained on natural images often struggle to extract high-level pathological features, such as specific textures and morphological structures. TransPath [30] trained a vision transformer-based feature encoder using a semantically relevant contrastive learning approach on a large number of WSIs. IBMIL [15] also employed a feature encoder that was pre-trained on nine pathological datasets using a Representations produced by these pathology-specific feature extractors significantly outperform those from feature encoders pre-trained on ImageNet [5] in downstream tasks.

The most common aggregation strategies for instance-based and embedding-based methods include pooling and attention mechanisms. Mean-MIL and Max-MIL aggregate representations then categorize through the average and maximum values respectively, but fixed aggregation mechanisms cannot adapt to varying inputs. In contrast, ABMIL employ attention mechanisms to aggregate features through trainable weights. Similarly, CLAM uses gate attention and a top-k selection strategy for bag label prediction. TransMIL, on the other hand, applies a linear approximation of self-attention to explore relationships between instances. WiKG [14] introduces a knowledge-aware attention mechanism, enhancing the capture of relative positional information among instances. HAT-Net+ [1] advances cell graph classification by leveraging a unique, parameter-free strategy to dynamically merge multiple hierarchical representations, effectively capturing the complex relationships and dependencies within cell graphs.

To enhance performance and stability, various methods employ data augmentation. For example, DTFD [34] increases the number of bags using a partitioning pseudo-bag split strategy. LNPL-MIL [22] enhances the robustness and generalization of MIL methods through feature augmentation.

Figure 2: Overall process: (a) the preprocess of WSI. (b) overall framework of AMD-MIL. (c) proposed mask denoise mechanism.

In terms of training strategies, IBMIL [15] utilizes interventional training to reduce the impact of contextual priors on relevance.Meanwhile, SSC-MIL [33] leverages semantic-similarity collaborative knowledge distillation to exploit latent bag information. MHIM-MIL [25] addresses key instances via hard example mining.

## 2.2 Approximate Self-Attention Mechanism

Self-attention mechanism is capable of grasping dependencies over long distances to facilitate comprehensive modeling. but its quadratic complexity limits the increase in input sequence length. Consequently, research on approximate self-attention mechanisms aims to approximate the complexity to linear without significantly compromising the global modeling capability.

Nystrom attention utilizes the nystrom method, a mathematical technique for estimating the eigenvalues and eigenvectors of large matrices. This method approximates self-attention mechanism by selecting a small subset of landmarks to represent the entire matrices, thereby reducing computational and storage requirements. Focused Linear [8] attention provides a nonlinear reweighting mechanism that can easily concentrate on important features. Agent attention introduces the concept of agents that represent key information within the input sequence. The computational complexity is significantly lowered by computing attention only among these agents instead of across the entire input sequence.

These advancements in approximate attention mechanisms provide a new perspective for improvements in the aggregator for MIL methods.

## 3 METHODOLOGY

### 3.1 MIL and Feature extraction

In the MIL methodology, each WSI is conceptualized as a labeled bag, wherein its constituent patches are considered as instances possessing indeterminate labels. Taking binary classification of WSIs as an example, the input WSI $X$ is divided into numerous patches $\{(x_1, y_1), \cdots, (x_N, y_N)\}$, encompassing $N$ instances of $x_i$. Under the MIL paradigm, the correlation between the bag's label, $Y$, and the labels of instances $y_i$ is established as follows:

$$Y = \begin{cases} 1, & \text{iff } \sum y_i > 0 \\ 0, & \text{else} \end{cases}. \tag{1}$$

Given the undisclosed nature of the labels for the instances $y_i$, the objective is to develop a classifier, $\mathcal{M}(X)$, tasked with estimating $\hat{Y}$. In alignment with methodologies prevalent in contemporary research, the classifier can be delineated into three steps: feature extraction, feature aggregation, and bag classification. All the processes can be defined as follows:

$$\hat{Y} \leftarrow \mathcal{M}(X) := h(g(f(X))), \tag{2}$$

where $f$, $g$, and $h$ represent the feature extractor, feature aggregator, and the MIL classifier.

The feature aggregator is considered to be the most important part of summarizing features, which can aggregate features of different patches. The attention mechanisms can discern the importance of patches in a WSI, and it is widely used in the feature aggregator.

Attention-based and self-attention-based MIL are the main methods currently used.

In the attention-based MIL [34], the feature aggregator can be defined as,

$$G = \sum_{i=1}^{N} a_i h_i = \sum_{i=1}^{N} a_i f(x_i) \in \mathbb{R}^D, \quad (3)$$

where $G$ is the bag representation, $h_i \in \mathbb{R}^D$ is the extracted feature for the patch $x_i$ through the feature extractor $f$, $a_i$ is the trainable scalar weight for $h_i$ and $D$ is the dimension of vector $G$ and $h_i$.

In the self-attention-based [27] MIL, the feature aggregator can be defined as,

$$Q = HW_Q, K = HW_K, V = HW_V, \quad (4)$$

$$O = \text{softmax}\left(\frac{QK^T}{\sqrt{d_q}}\right)V = SV, \quad (5)$$

where $W_Q$, $W_K$, and $W_V$ represent trainable matrices, $H$ denotes the collection of patch features, and $O$ has integrated the attributes of the other features.

## 3.2 Attention Aggregator

During the computation of $\text{Sim}(Q, K)$ as defined in Eq. 5, the algorithmic complexity scales quadratically with $O(N^2)$. Given that $N$ frequently comprises several thousand elements, this substantially extends the expected computational time. Linear attention offers a reduction in computational time but at the expense of information. To mitigate this issue, transmil [21] employs nystrom approximation for Eq. 5 [32]. The matrices $\tilde{Q}$ and $\tilde{K}$ are constructed, and the mean of each segment is computed as follows:

$$\tilde{Q} = [\tilde{q}_1; \ldots; \tilde{q}_m], \quad \tilde{q}_j = \frac{1}{m}\sum_{i=(j-1)\times l+1}^{(j-1)\times l+m} q_i, \quad \forall j = 1, \ldots, m \quad (6)$$

$$\tilde{K} = [\tilde{k}_1; \ldots; \tilde{k}_m], \quad \tilde{k}_j = \frac{1}{m}\sum_{i=(j-1)\times l+1}^{(j-1)\times l+m} k_i, \quad \forall j = 1, \ldots, m \quad (7)$$

where $\tilde{Q} \in \mathbb{R}^{m \times D}$ and $\tilde{K} \in \mathbb{R}^{m \times D}$.

The approximation of the $\hat{S}$ in Eq. 5 can then be expressed as:

$$\hat{S} = \text{softmax}\left(\frac{Q\tilde{K}^T}{\sqrt{d_q}}\right)Z^*\text{softmax}\left(\frac{\tilde{Q}K^T}{\sqrt{d_q}}\right), \quad (8)$$

where, $Z^*$ represents the approximate solution to $z(\tilde{Q}, \tilde{K}, Z) = 0$, necessitating a linear number of iterations for convergence.

In MIL tasks, nystrom attention filters out patches with important features because of the filtering of patches. Moreover, the difference in N will lead to an overall imbalance during local down-sampling. So we consider agent attention methods with linear time complexity and the agent attention mechanism [9] can be written as:

$$O = \sigma(QA^T)\sigma(AK^T)V, \quad (9)$$

where $\sigma(\cdot)$ is the Softmax function, $Q, K, V$ are defined in equation Eq. 4. Here $A \in \mathbb{R}^{n \times D}$ is the agent matrix pooling from $Q$. The

---

**Algorithm 1** Agent Aggregator With Mask Denoise Mechanism

**Input:** H : ( B , N , D )
**Output:** Y : ( B , N , D )
1: // H : bag features
2: // B : batch    N : token length    D : feature dimensions
3: $Q, K, V$ : ( B , N , D ) ⟵ nn.linear ( H )
4: $AGENT$ : ( B , n , D ) ⟵ trainable parameters
5: // n : number of agent tokens
6: $Q_A$ : ( B , N , n ) ⟵ torch.matmul ( $Q$ , $AGENT^T$ )
7: $K_A$ : ( B , n , N ) ⟵ torch.matmul ( $AGENT$ , $K^T$ )
8: $V_A$ : ( B , n , D ) ⟵ torch.matmul ( $K_A$ , $V$ )
9: $MASK$ : ( B , n , D ) ⟵ nn.linear ( $V_A$ )
10: $THR$ : ( B , 1 ) ⟵ nn.linear ( $V_A$ ) . suqeeze ( ) . mean ( -1 )
11: $MASK_t$ : ( B , n , D) ⟵ torch.where ( $MASK > THR$ , 1 , 0 )
12: $V_M$ : ( B , n , D ) ⟵ torch.mul ( $V_A$ , $MASK_t$ )
13: $DN$ : ( B , n , D ) ⟵ nn.linaer ( $V_A$ )
14: $V_{MD}$ : ( B , n , D ) ⟵ torch.add ( $V_M$ , $DN$ )
15: $Y$ : ( B , N , D ) ⟵ torch.matmul( $Q_A$ , $V_{MD}$ )
16: // Y : weighted fbag features
17: **return** $Y$

---

term $D$ stands for the feature dimension, while $n$ refers to the agent dimension and acts as a hyperparameter.

Given that the agent is non-trainable and the distribution of attention scores may not be optimal, it becomes imperative to establish an adaptive agent capable of dynamically adjusting the attention score distribution to enhance model performance and flexibility.

## 3.3 Agent Mask Denoising Mechanism

As illustrated in Figure 1, our overall framework is based on Eq. 5 and Eq. 9. Proposed Overall framework is in Figure 2. Before the input features are processed by the model, a class token is embedded into them, resulting in the feature matrix $H \in \mathbb{R}^{D \times (N+1)}$, where $D$ is the dimension of the features and $(N + 1)$ represents the number of patches, with the accounting for the embedded class token.

**Trainable Agent.** In the methodology outlined earlier, matrix $A$ in Eq. 9 is initially from matrix $Q$ through mean pooling, $A = pooling(Q) \in \mathbb{R}^{n \times D}$, indicating a limitation in encapsulating the entirety of information present within $Q$. To overcome this limitation, $A$ is defined as a trainable matrix. Through matrix $A \in \mathbb{R}^{n \times D}$, the intermediate matrices $Q_A = QA^T \in \mathbb{R}^{(N+1) \times n}$ and $K_A = AK^T \in \mathbb{R}^{n \times (N+1)}$ can be obtained. Utilizing the general attention strategy, the intermediate variable is

$$V_A = \sigma(K_A)V$$
$$= \sigma(AK^T)V \in \mathbb{R}^{n \times D}. \quad (10)$$

**Mask Agent.** In this MIL task, most regions of a WSI do not contribute much to the prediction, so a learnable mask is generated by using the trainable threshold to mask the information

$$\tau = \sigma(p(W_\tau V_A^T)), \quad (11)$$

where $W_\tau \in \mathbb{R}^{1 \times D}$, function $p$ is an adjustable aggregate function such as mean-pooling, and $\tau$ is the threshold.

**Table 1: Performance of AMD-MIL on CAMELYON-16, CAMELYON-17, TCGA-LUNG, and TCGA-KIDNEY datasets.**

| Method | CAMELYON-16 | | | CAMELYON-17 | | | TCGA-LUNG | | | TCGA-KIDNEY | | |
|---|---|---|---|---|---|---|---|---|---|---|---|---|
| | ACC(%) | AUC(%) | F1(%) | ACC(%) | AUC(%) | F1(%) | ACC(%) | AUC(%) | F1(%) | ACC(%) | AUC(%) | F1(%) |
| MeanMIL | $79.4_{2.12}$ | $83.3_{2.31}$ | $78.5_{2.23}$ | $69.5_{2.14}$ | $69.2_{1.51}$ | $65.4_{2.21}$ | $82.4_{1.31}$ | $86.4_{1.62}$ | $82.0_{2.11}$ | $90.3_{1.49}$ | $93.1_{1.04}$ | $87.9_{1.10}$ |
| MaxMIL | $76.4_{0.91}$ | $80.4_{2.04}$ | $75.4_{1.55}$ | $66.7_{1.45}$ | $70.2_{1.52}$ | $65.8_{0.92}$ | $87.7_{1.12}$ | $87.4_{1.34}$ | $88.7_{1.72}$ | $91.2_{1.58}$ | $93.5_{1.13}$ | $86.8_{1.32}$ |
| ABMIL [11] | $84.8_{1.14}$ | $85.9_{1.05}$ | $84.1_{1.22}$ | $78.7_{1.98}$ | $77.3_{1.64}$ | $75.3_{1.32}$ | $88.4_{2.04}$ | $93.1_{2.23}$ | $87.6_{2.10}$ | $91.6_{0.93}$ | $94.1_{0.82}$ | $88.5_{1.23}$ |
| G-ABMIL [11] | $84.0_{1.26}$ | $85.3_{1.11}$ | $83.6_{1.34}$ | $79.9_{1.76}$ | $79.3_{1.87}$ | $76.2_{1.82}$ | $87.6_{1.77}$ | $91.0_{1.63}$ | $86.3_{1.82}$ | $91.4_{1.15}$ | $93.8_{1.04}$ | $89.4_{1.20}$ |
| CLAM-MB[17] | $91.1_{0.82}$ | $94.5_{0.78}$ | $90.7_{0.91}$ | $83.6_{1.42}$ | $84.8_{0.71}$ | $81.3_{1.70}$ | $89.3_{1.23}$ | $94.2_{1.18}$ | $88.2_{1.42}$ | $91.2_{0.78}$ | $92.9_{0.66}$ | $90.2_{0.74}$ |
| CLAM-SB[17] | $91.9_{1.58}$ | $94.3_{1.27}$ | $91.1_{1.54}$ | $83.9_{1.48}$ | $85.2_{1.64}$ | $81.5_{1.44}$ | $87.3_{1.23}$ | $93.1_{1.41}$ | $89.1_{1.64}$ | $89.7_{1.76}$ | $93.9_{1.67}$ | $90.2_{1.98}$ |
| DSMIL [13] | $85.8_{0.63}$ | $91.8_{0.72}$ | $86.2_{0.75}$ | $72.2_{0.76}$ | $72.8_{0.86}$ | $72.4_{0.72}$ | $85.2_{0.85}$ | $93.6_{0.82}$ | $85.9_{0.94}$ | $90.2_{0.78}$ | $94.7_{0.66}$ | $86.2_{0.71}$ |
| TransMIL [21] | $87.8_{3.24}$ | $93.7_{3.21}$ | $88.7_{3.61}$ | $75.4_{4.02}$ | $74.6_{3.77}$ | $71.7_{3.23}$ | $87.9_{3.22}$ | $94.1_{3.12}$ | $88.2_{3.40}$ | $91.1_{2.56}$ | $92.5_{2.75}$ | $89.3_{2.98}$ |
| DTFD [34] | $89.4_{0.73}$ | $92.3_{0.92}$ | $88.4_{0.78}$ | $76.3_{0.67}$ | $77.8_{0.88}$ | $75.4_{0.82}$ | $86.8_{1.04}$ | $94.7_{0.75}$ | $86.1_{0.91}$ | $91.5_{0.79}$ | $95.3_{0.85}$ | $90.8_{0.77}$ |
| RRT [26] | $88.6_{0.96}$ | $93.4_{1.23}$ | $89.9_{0.94}$ | $76.7_{2.14}$ | $74.5_{1.49}$ | $72.3_{1.22}$ | $87.1_{2.10}$ | $93.2_{1.87}$ | $88.0_{1.72}$ | $92.8_{1.46}$ | $93.6_{1.68}$ | $90.7_{1.38}$ |
| WiKG [14] | $91.1_{1.26}$ | $94.6_{1.20}$ | $90.8_{1.15}$ | $80.3_{1.41}$ | $80.4_{1.38}$ | $77.8_{1.20}$ | $89.7_{0.96}$ | $94.6_{0.72}$ | $89.3_{1.23}$ | $93.2_{1.11}$ | $95.9_{0.84}$ | $91.6_{1.12}$ |
| **AMD-MIL** | $\mathbf{92.9_{2.73}}$ | $\mathbf{96.4_{2.89}}$ | $\mathbf{92.7_{2.83}}$ | $\mathbf{85.0_{1.32}}$ | $\mathbf{85.3_{0.69}}$ | $\mathbf{82.7_{1.24}}$ | $\mathbf{90.5_{1.51}}$ | $\mathbf{95.2_{0.70}}$ | $\mathbf{90.5_{1.59}}$ | $\mathbf{94.4_{1.13}}$ | $\mathbf{97.3_{0.74}}$ | $\mathbf{92.9_{1.01}}$ |

Calculate the importance of each feature to optimize the important features in the hidden space. The selection of features will have the risk of information loss. To balance important information selection and the original characteristics of the aggregation, we proposed a new module which can be defined as,

$$V_{MD_{ij}} = V_{A_{ij}} \mathbb{I}_{M_{ij} > \tau} + DM_{ij}, \qquad (12)$$

where $M = W_M V_A$ is the threshold matrix to obtain the importance of each feature, and $DN = W_{DN} V_A$ is the denoise matrix to aggregate information.

**Agent Visualization.** The foundational agent attention architecture lacks the capability to produce a variable concentration score for sequences. To address this limitation, we outline a methodology that facilitates the visualization of attention scores:

$$Att_i = \sum_{j=1}^{n} Q_{A_{0,j}} K_{A_{j,i+1}}, \qquad (13)$$

where $Att_i$ is the attention score of the feature $h_i$.

**AMD.** Establishing the aforementioned modules, we introduce a novel framework titled *Attention Mask Denoising*. This framework, as illustrated in the Figure 2, encompasses a learning-based agent attention mechanism, feature selection, and feature aggregation within the hidden space. The algorithm process is shown in Algorithms 1 and the module can be expressed as:

$$O = \sigma(QA^T)\sigma(AK^T)V$$
$$= \sigma(QA^T)V_{MD}, \qquad (14)$$

where $V_{MD}$ represents the mask and denoising module Eq. 12.

Due to the difference in the threshold selection method, the other two feature threshold selection strategies are considered as follows:

- **Mean-AMD**. Mean selection: selected the average value in the features as the threshold selected by all features.
- **CNN-AMD**. CNN selection: through the method of group convolution, the characteristics of different groups are reduced, and the average value between the groups is the threshold.

## 4 EXPERIMENTS

### 4.1 Datasets and Evaluation Metrics

In our study, we employ four datasets available in the public domain to assess the performance of the approach we proposed.

**CAMELYON-16** is a dataset for early-stage breast cancer lymph node metastasis detection. The dataset comprises 399 WSIs, which are officially split into 270 for training and 129 for testing. Following the split, we employed 6-fold cross-validation to ensure that all data were utilized for both training and testing, thereby preventing overfitting to the official test set. In addition, we employ the pre-trained weights from CAMELYON-16 dataset to perform inference on the external dataset **CAMELYON-17** only once. Subsequently, we report both the mean and variance of the evaluation metrics.

**TCGA-LUNG** comprises 1034 WSIs, encompassing 528 WSIs from Lung Adenocarcinoma (LUAD) cases and 507 WSIs from Lung Squamous Cell Carcinoma (LUSC) cases. We randomly split the dataset into training, validation, and testing sets with a ratio of 65:10:25. 4-fold cross-validation is used, and the mean and standard deviation of performance metrics are reported.

**TCGA-KIDNEY** comprises 1075 WSIs, encompassing 117 WSIs from Kidney Chromophobe (KICH) cases, 539 WSIs from Kidney Clear Cell Carcinoma (KIRC) cases, and 419 WSIs from Kidney Papillary Cell Carcinoma (KIRP) cases. We randomly split the dataset into training, validation, and testing sets with a ratio of 65:10:25. We adopt 4-fold cross-validation and report the mean and standard deviation of evaluation metrics.

We report the evaluation metrics as the mean and standard deviation of the macro F1 score, the area under the curve (AUC) for one-versus-rest scenarios, and the slide-level accuracy (ACC).

### 4.2 Implementation Details

During the preprocessing phase, we generated non-overlapping patches of 256x256 pixels at 20x magnification for the datasets CAMELYON-16, CAMELYON-17, TCGA-KIDNEY, and TCGA-LUNG. This procedure yielded an average count of approximately 9024, 7987, 13266, and 10141 patches per bag for the respective datasets.

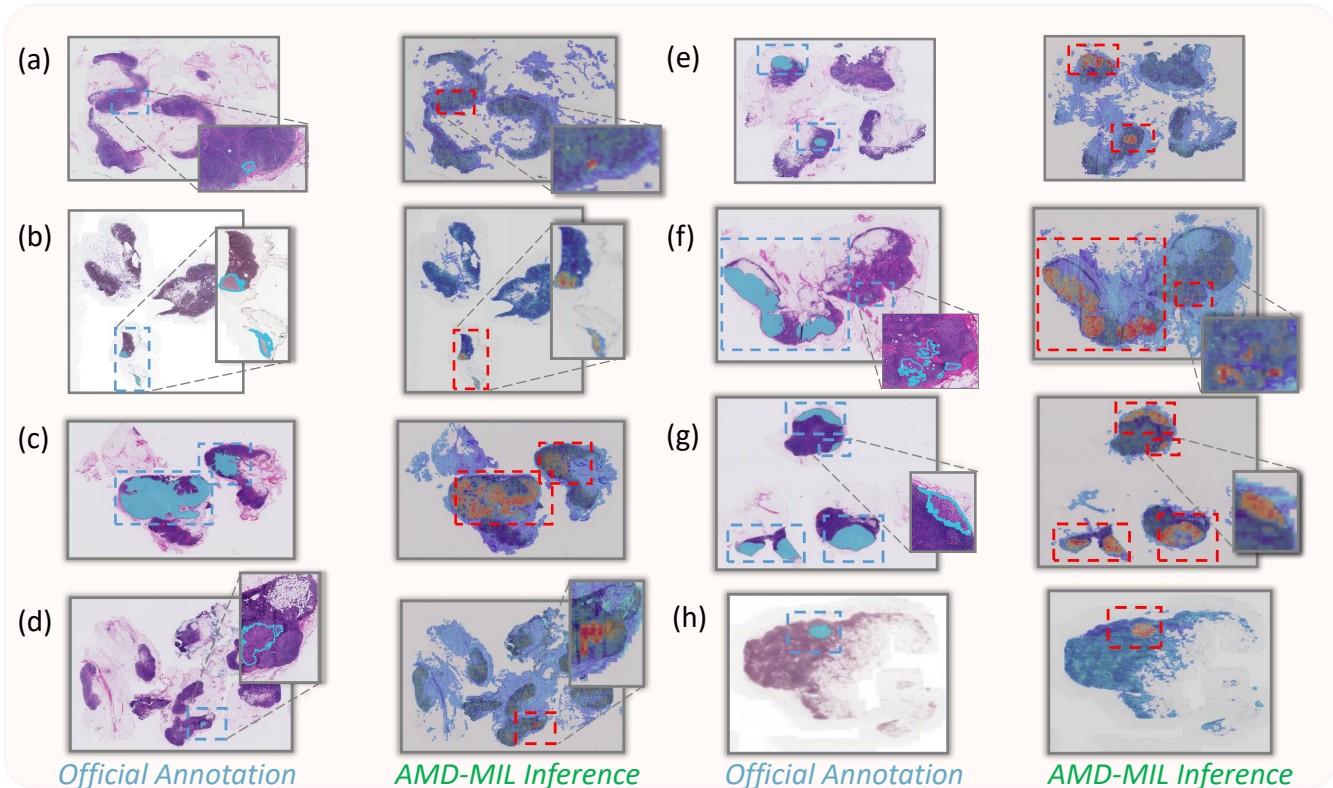

**Figure 3: Visualization of AMD-MIL Attention Distribution Compared to Official Annotations on CAMELYON-16.**

Uniform hyperparameters were maintained across all experiments. Each experiment was conducted on a workstation equipped with NVIDIA RTX A100 GPUs, utilizing ImageNet [5] pre-trained ResNet50 [10] as the feature encoding model. The Adam optimization algorithm was used, incorporating a weight decay of 1e-5. The initial learning rate was set at 2e-4, and cross-entropy loss was employed as the loss function.

For ADM-MIL, we introduce a hyperparameter, agent numbers, to control the number of agent tokens.

## 4.3 Comparison with State-of-the-Art Methods

In this study, we present the experimental results of our newly developed AMD-MIL framework applied to the CAMELYON-16, CAMELYON-17, TCGA-LUNG, and TCGA-KIDNEY datasets. We compared this framework with various methodologies, including MeanMIL, MaxMIL, ABMIL [11], CLAM [17], DSMIL [13], TransMIL [21], DTFD [34], RRT [26], and WiKG [14], to evaluate its effectiveness.

As shown in Table 1, the AMD-MIL framework demonstrated superior performance, achieving AUC scores of 96.4% for CAMELYON-16, 85.3% for CAMELYON-17, 95.2% for TCGA-LUNG, and 97.3% for TCGA-KIDNEY. Notably, these scores consistently exceeded those of the previously mentioned comparative methods, highlighting the framework's exceptional ability to dynamically adapt to inputs. This adaptability enables the effective capture of key features, accurately representing the original bag features.

As demonstrated in Table 3, We also conducted a comparative analysis to evaluate the impact of different threshold selection methods on the metrics. We found that using a linear layer for aggregation outperforms both average pooling and group-convolution.

## 4.4 Interpretability Analysis

We conducted an interpretability analysis of AMD-MIL. In Figure 3, the blue-masked areas denote the official annotations of cancerous regions in the CAMELYON-16 dataset, whereas the heatmap regions represent the distribution of agent attention scores across all patches constituting the WSIs, calculated according to Eq. 13. The attention scores indicate the contribution level of instances to the classification outcome, and it is distinctly observable that areas of high attention scores align closely with the annotated cancerous regions. This demonstrates that the AMD-MIL classification relies on the cancerous ROI, mirroring the diagnostic process of pathologists, thereby providing substantial interpretability for clinical applications. AMD-MIL not only possesses robust localization capabilities for macro-metastases but also accurately focuses on micro-metastases. For example, Figure 3 (f), which includes both macro and micro-metastases, AMD-MIL can also concurrently localize to different areas.

## 4.5 Ablation Study

**Effectiveness of Agent Aggregator.** The trainable agent aggregator employs agent tokens as intermediate variables for the query

**Table 2: Comparison between TransMIL with AMD-MIL and the effectiveness of the components of AMD-MIL.**

| Dataset | Component | | | | | ACC(%) | AUC(%) | F1(%) |
|---------|---------|-------|-------|------|---------|--------|--------|-------|
|         | nystrom | agent | train | mask | denoise |        |        |       |
| CAMELYON-16 | ✓ |   |   |   |   | $87.8_{3.24}$ | $93.7_{3.21}$ | $88.7_{3.61}$ |
|         |   | ✓ |   |   |   | $89.3_{2.90}$ | $93.8_{3.08}$ | $88.6_{3.09}$ |
|         |   | ✓ | ✓ |   |   | $91.5_{3.62}$ | $95.6_{3.06}$ | $91.2_{3.67}$ |
|         |   | ✓ | ✓ | ✓ |   | $93.0_{2.72}$ | $96.0_{3.00}$ | $92.7_{2.80}$ |
|         |   | ✓ | ✓ | ✓ | ✓ | $\mathbf{92.9_{2.73}}$ | $\mathbf{96.4_{2.89}}$ | $\mathbf{92.7_{2.83}}$ |
| LUNG | ✓ |   |   |   |   | $87.9_{3.22}$ | $94.1_{3.12}$ | $88.2_{3.40}$ |
|         |   | ✓ |   |   |   | $88.4_{0.86}$ | $93.5_{0.86}$ | $88.4_{0.88}$ |
|         |   | ✓ | ✓ |   |   | $87.5_{1.00}$ | $92.6_{3.47}$ | $87.4_{1.02}$ |
|         |   | ✓ | ✓ | ✓ |   | $90.2_{1.19}$ | $94.6_{0.91}$ | $90.2_{1.19}$ |
|         |   | ✓ | ✓ | ✓ | ✓ | $\mathbf{90.5_{1.51}}$ | $\mathbf{95.2_{0.70}}$ | $\mathbf{90.5_{1.59}}$ |
| KIDNEY | ✓ |   |   |   |   | $91.1_{2.56}$ | $92.5_{2.75}$ | $89.3_{2.98}$ |
|         |   | ✓ |   |   |   | $93.7_{0.43}$ | $97.0_{0.57}$ | $91.1_{0.94}$ |
|         |   | ✓ | ✓ |   |   | $93.7_{1.13}$ | $\mathbf{97.7_{0.57}}$ | $91.4_{0.18}$ |
|         |   | ✓ | ✓ | ✓ |   | $93.4_{1.06}$ | $97.6_{0.57}$ | $90.7_{0.13}$ |
|         |   | ✓ | ✓ | ✓ | ✓ | $\mathbf{94.4_{1.13}}$ | $97.3_{0.74}$ | $\mathbf{92.9_{1.01}}$ |

**Table 3: Different thresh select methods on Camelyon-16.**

| Thresh | ACC(%) | AUC(%) | F1(%) |
|--------|--------|--------|-------|
| Mean | $91.3_{2.12}$ | $96.0_{2.21}$ | $91.0_{3.83}$ |
| CNN | $91.2_{3.66}$ | $95.8_{3.39}$ | $91.0_{3.69}$ |
| Linear | $92.9_{2.73}$ | $96.4_{2.89}$ | $92.7_{2.83}$ |

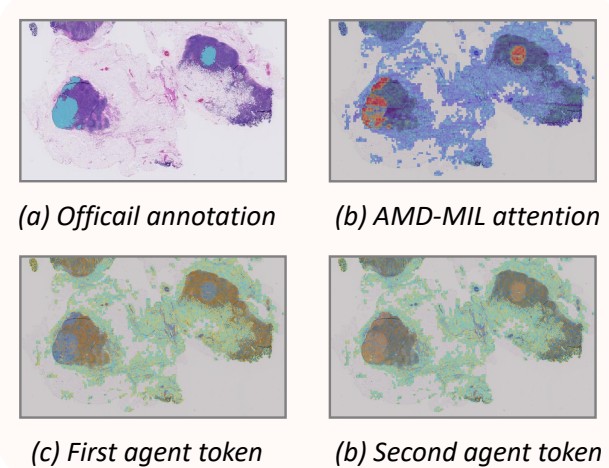

*(a) Officail annotation*  *(b) AMD-MIL attention*

*(c) First agent token*  *(b) Second agent token*

**Figure 4: Attention distribution of different agent tokens.**

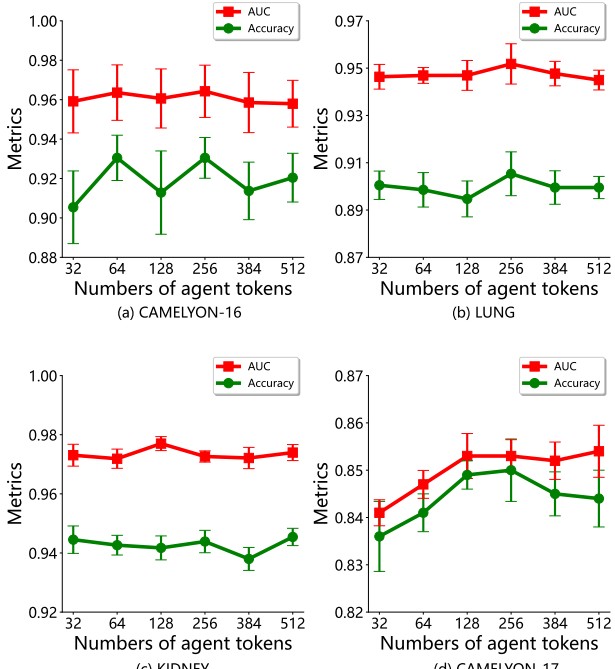

**Figure 5: Influence of the number of agent tokens**

and key of original self-attention mechanism, ensuring global modeling while approximating linear attention. We compare the trainable-agent aggregator with the original pooling-agent aggregator and the nystrom attention aggregator from TransMIL. The original pooling-agent aggregator reduces parameter count via a proxy mechanism and achieves enhanced global modeling through a broadcasting mechanism. As shown in Table 2, it significantly outperforms the

nystrom attention aggregator from TransMIL across three datasets. However, the pooling-agent aggregator struggles to dynamically adapt to inputs, and its pooling mechanism may average out important instances. By initializing the agent as a trainable parameter, we observe a relative improvement in metrics compared to the pooling-agent aggregator.

We further explored the attention distribution patterns among various agent tokens, as depicted in Figure 4. Notably, the first agent

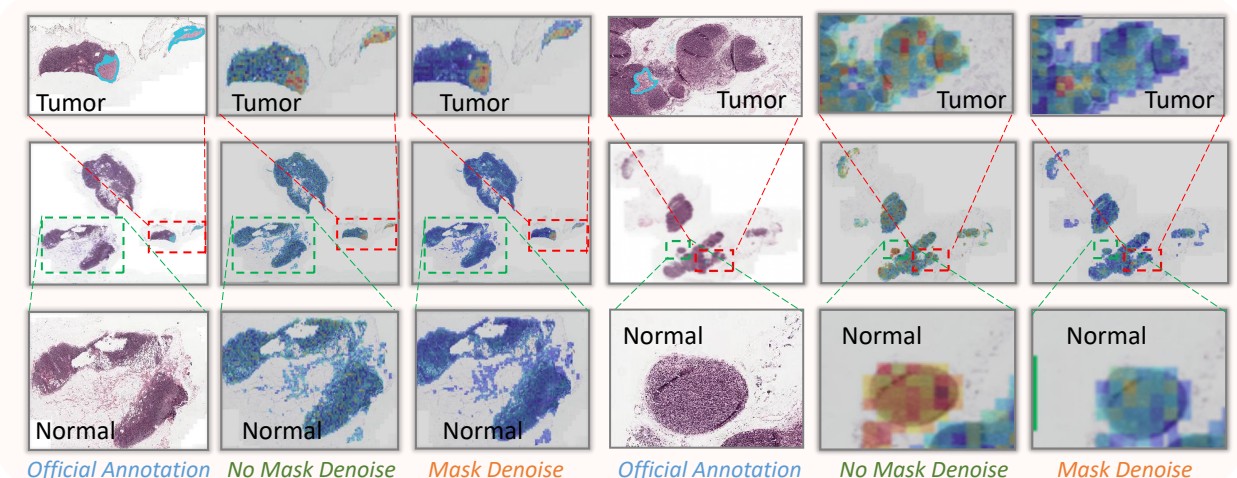

Figure 6: Effectiveness of the mask denoise mechanism.

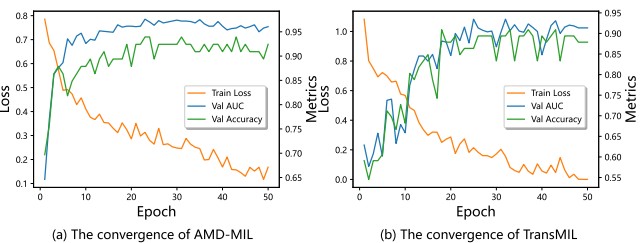

(a) The convergence of AMD-MIL        (b) The convergence of TransMIL

Figure 7: Model convergence of AMD-MIL and TransMIL.

token is oriented towards non-cancerous tissues, whereas the second agent token is aligned with cancerous zones. This observation underscores the fact that different agent tokens exhibit unique focal points across the spectrum. This variance in attention ensures that during the broadcasting process by agents, diverse queries are able to concentrate on their respective areas of significance, thereby enhancing the model's capacity to differentiate between critical and non-critical regions effectively.

Additionally, the number of agent tokens constitutes a critical hyperparameter in AMD-MIL. As shown in Figure 5, comprehensive experiments were conducted on four datasets with agent token counts of 32, 64, 128, 256, 384, and 512. We observed fluctuations in the ACC on the CAMELYON-16 dataset as the number of agent tokens increased, possibly due to the dataset's small size causing instability. On the CAMELYON-17 dataset, the AUC increases with the number of agents, while the ACC shows a trend of first increasing and then decreasing. Additionally, the AUC on CAMELYON-16 dataset, alongside the AUC and ACC for TCGA datasets, maintained relatively stable performance. This consistency aligns with the results of experiments modifying the number of agent tokens within shallow attention stacks on natural images [9].

**Effectiveness of Mask Denoise Mechanism.** The mask denoise mechanism enables a more rational allocation of attention scores by dynamically masking out representations of less significant instances. Denoising matrices are used to mitigate the noise introduced during the mask process. As shown in Table 2, we compare the metrics of the agent aggregator with and without the mask and denoise mechanisms. On average, the mask denoise mechanism enhances performance metrics. Figure 6 contrasts the distribution of instance attention scores for agent aggregators with and without the mask denoise mechanism. It is evident that, even without the mask denoise mechanism, some WSIs could be correctly classified. However, a portion of the higher attention is allocated to non-cancerous areas, diminishing the model's interpretability. For micro-metastatic cancer as shown in Figure 6 (b), such bias could lead to erroneous classification results, posing challenges for clinical application. With the incorporation of the mask denoise mechanism, the distribution of attention scores becomes more concentrated on the ROIs within cancerous areas, granting significantly lower attention to non-cancerous regions. This suggests that the mask denoise mechanism can dynamically correct attention scores to achieve improved interpretability.

## 5 CONCLUSION

In pathological image analysis, using attention-based aggregators significantly advances MIL methods. However, traditional attention mechanisms, due to their quadratic complexity, struggle with processing high-resolution images. Additionally, approximate linear self-attention mechanisms also have inherent limitations. To address these challenges, we introduce AMD-MIL, a novel approach for dynamic agent aggregation and feature refinement. Our validation on three distinct datasets not only demonstrates AMD-MIL's effectiveness but also its ability for instance-level interpretability. Moving forward, we aim to further evaluate its robustness across a broader range of datasets and explore its potential clinical applications.

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
