# OpenReview forum: "Agent Aggregator with Mask Denoise Mechanism for Histopathology Whole Slide Image Analysis"
_acmmm.org/ACMMM/2024/Conference — MM2024 Poster_

### Official Review · Reviewer_ziCK · 2024-05-05

**Rating:** 2
**Confidence:** 4

**Summary:**

The paper proposes an agent aggregator with a mask denoising mechanism for multiple instance learning. The agent token acts as an intermediary variable between the query and key, facilitating the implicit calculation of instance significance. The newly introduced mask denoising mechanism aims at dynamically refining attention scores through the adjustment of instance representations.

The paper demonstrates the effectiveness of the proposed method by comparing it with the latest representative methods on four datasets.

**Strengths:**

1. The theoretical basis for the design of the method is substantial.

2. The experiments are quite comprehensive, providing evidence of the effectiveness of the proposed method from multiple perspectives.

**Limitations:**

1. The description of the methodology (Sec. 3) is somewhat difficult to follow. It needs to be aligned with the method framework depicted in Fig. 2. More symbol annotations should be added to Fig. 2.

2. The description and the presented figures in each section of the paper are rough. The issues include but are not limited to: (1) $\textbf{agent}$ is misspelled in the caption of Fig. 1. (2) In the para. 2 of Sec. 2.1, the sentence before "Representations" is incomplete. (3) In the penultimate paragraph of Sec. 3.2, it's unclear which formula "D" and "n" refer to. (4) In Sec. 3.3, $\textbf{Overall}$$\rightarrow$$\textbf{overall}$. (5) Fig. 3 and 6 is quite rough. (6) In Fig. 6, there are no labels assigned to each set of images, such as assigning "a", "b", etc. However, in the text, "Fig. 6(b)" is used.

I believe this is a meaningful and comprehensive work, but the descriptions and figures in the paper require significant improvement.

**Suitability:**

2

---

### Official Review · Reviewer_qvzj · 2024-05-09

**Rating:** 4
**Confidence:** 3

**Summary:**

In this work, the authors proposed to study the WSI classification task based on MIL. Specifically, the proposed method is for the attention mechanism, by proposing a learnable agent token, as well as the denoise mechanism based on the learnable masks. Extensive experiments on various WSI classification studies have indicated the effectiveness of the proposed method. Ablation studies have been presented to prove the effectiveness of each module in the overall framework.

**Strengths:**

1) The proposed method has alleviated the limitations in the existing agent attention mechanism [9], by making the agent learnable. In addition, the mask modules are further designed to refine the features of the agent.

2) Comparison experiments have indicated the effectiveness of the proposed method by outperforming various comparison works for WSI classification.

3) Ablation studies have been presented to indicate the effectiveness of the proposed modules.

4) The source code is provided during the paper review process.

**Limitations:**

1) The proposed method has not been validated on other applications. In this work, the major contribution is the agent attention module, which is not restricted by some specific domain knowledge in the digital pathology domain. Therefore, they can be easily extended to other applications based on attention modules (e.g., [9]). Such validations are also important to evaluate the performance of the overall work.

2) The computational complexity analysis on each proposed mechanism is missing. Although the proposed modules can introduce performance gain, it is not clear whether they are also efficient.

**Suitability:**

2

---

### Official Review · Reviewer_eCnA · 2024-05-11

**Rating:** 5
**Confidence:** 3

**Summary:**

The utilization of attention-based aggregators in pathological image analysis substantially enhances MIL methods. However, due to their quadratic complexity, conventional attention mechanisms encounter difficulties in processing high-resolution images. Furthermore, approximate linear self-attention mechanisms possess inherent restrictions as well. The authors proposed AMD-MIL, an innovative, dynamic agent aggregation and feature refinement method, to address these difficulties. Their validation on three distinct datasets demonstrates the efficacy of AMD-MIL and its interpretability at the instance level.

**Strengths:**

The paper is flawless in clarity, presentation, and structure. The research motivation is appealing, and the abstract captures the study's substance. This approach also has the potential to advance Whole Slide Image (WSI) analysis studies and increase the literature in this sector.

**Limitations:**

1- Don't you think the Classification head is missing a link?
2- How were the results from the existing methods references [Table 1] computed? Were the methods re-implemented, or was the performance directly reported from their papers? If the results were taken from their original papers, was it ensured that those methods used the same train-validation and test split compared to this work, the same dataset? (This is very important to me for convincing)
3- The lock symbol typically conveys security or encryption; why do you employ such a symbol in your Figure 2?
4- Performance is crucial, but validating the proposed approach's time efficiency would be beneficial. Don't you think adding this as a motivation would increase the suitability of your method?
5- The authors mistakenly missed the data split ratio for CAMELYON-16 and CAMELYON-17.
6- Please explain why different fold sizes (e.g., 6 and 4) were used in the experimentation. While justification for the choice of fold sizes would be helpful, providing additional context or rationale could further enhance the clarity and reproducibility of the experimental setup.
7- "We generated non-overlapping patches of 256x256." However, given that they are non-overlapping, there's a possibility of missing important information between adjacent patches. Could you please elaborate on how data coverage and continuity are ensured across the entire image despite using non-overlapping patches?
8—Please explain more about the first and second agent tokens in Figure 4. In contrast, Figure 5 visualizes 32 to 512 tokens. How about aiding an additional image, considering the best/average/worst token number performer, and presenting an image to compare your method on four datasets to comprehend your analysis?

**Suitability:**

3

---

### Official Review · Reviewer_Lrmz · 2024-05-17

**Rating:** 4
**Confidence:** 3

**Summary:**

The paper proposes a multi-instance-learning-based WSI classification framework. In this framework, the authors propose an agent attention mechanism with learnable agent tokens to address the computational complexity of self-attention. Additionally, the authors propose a mask denoising mechanism to reduce the attention score of unimportant regions of a slide. The experimental results demonstrate that the framework achieves state-of-the-art performance on four WSI classification datasets.

**Strengths:**

1. The learnable agent attention is useful.

2. The experiments are substantial, and the WSI classification performance is good.

3. The code is provided.

**Limitations:**

1. The motivation to use agent attention is to reduce computational complexity. However, the motivation for the agent mask denoising mechanism is not very convincing. If most regions are not useful, why not select only the regions with the highest attention scores so that the model can learn to identify and prioritize useful regions? Why is there a need to learn an additional mask?

2. The classification results of CVPR2024-RRT on the Camelyon16 dataset are much lower than those reported in its paper. Why is this the case? Why do you use 6-fold for the Camelyon dataset? I notice that most previous methods use 10-fold.

3. There is a lack of comparison of attention distribution visualization with other attention-based methods.

4. The standard deviations of AMD-MIL on the Camelyon16 dataset are much higher than those of existing methods.

5. The application or method is not quite related to this conference.

6. The manuscript lacks clarity, making it difficult to follow. For example:

[1] There is a spelling error in Figure 1 (b) - 'agnets’

[2] There is no reference to 'Agent attention’ in Section 2.2, and there are insufficient references for the self-attention mechanism.

[3] In Equation (5), the term 'd_q' is not introduced.

[4] In Section 3.2, Lines 396-397, the sentence 'Nystrom attention filters out patches with important features because of the filtering of patches.’ is unclear. What does this mean?

[5] In Equation (12), I suspect 'DM' should be 'DN'. The meaning of the notation 'Ι’, 'W_M', and 'W_DN' is missing.

[6] In equation (11), what is 'W_𝜏' and `𝜏' is a threshold for what usage?

[7] In Algorithm 1, the variables like 'MASK', 'MASK_t', 'V_M' and many others are not appeared and explained in the manuscript.

**Suitability:**

1

---

### Meta-Review · Area_Chair_xcVg · 2024-07-04

**Recommendation:** Accept (Poster)
**Confidence:** 3

**Metareview:**

The paper addresses the challenging task of accurate classification of whole slide images (WSIs) in histopathology analysis, crucial for enhancing clinical diagnosis. By leveraging multiple instance learning (MIL) within a weakly supervised framework, the proposed AMD-MIL approach introduces an innovative agent aggregator with a mask denoise mechanism. This mechanism effectively addresses limitations of existing attention-based strategies by dynamically adjusting instance importance and mitigating computational complexities associated with self-attention.

Key strengths of AMD-MIL include its ability to sensitively capture micro-metastases in cancer through improved attention allocation and enhanced interpretability. The learnable matrices for mask denoising contribute to refining feature representations, thereby achieving superior classification performance across CAMELYON-16, CAMELYON-17, TCGA-KIDNEY, and TCGA-LUNG datasets. The extensive experimental validation demonstrates significant advancements over state-of-the-art methods, highlighting AMD-MIL as a promising approach for WSI classification and localization tasks in histopathology.

Overall, the paper's rigorous methodology, comprehensive experimentation, and focus on clinical applicability position AMD-MIL as a valuable contribution to advancing the field of histopathology analysis.